# Estimation of Tritium Concentration in the Rain- and Groundwater in the Dry River of Hafr Al Batin, Saudi Arabia

**Al Mamun**

Department of Physics, College of Science, University of Hafr Al Batin, Al Jamiah, Hafr Al Batin 39524, Saudi Arabia; almamun@uhb.edu.sa; Tel.: +966-055-705-8027

**Abstract:** Natural tritium is widely dispersed in the environment, leading to human exposure to tritium through water ingestion, inhalation of tritium gas, and absorption through the skin. High levels of tritium in the environment can pose significant health risks due to the potential carcinogenic effects of tritium. Therefore, this study aims to estimate the tritium concentration in spiked water, rainwater, and groundwater by a liquid scintillation counter (LSC). Due to the lower detection levels of the LSC, an enrichment process of tritium ten- to thirtyfold was carried out using the electrolysis process. The enriched water was then analyzed to estimate the tritium concentration. Five reference samples were prepared by diluting a standard with known concentrations (spiked sample) to verify the precision of the measurement technique. The estimated tritium levels of the rainwater and the groundwater were within the 0.3 to 3.5 tritium units (TU) range. A comparatively higher tritium concentration was estimated for the shallow wells in the lower valley of the monitored areas. However, relatively lower concentrations were detected for wells located in the hillside areas. These findings will assist regulatory authorities in formulating strategies to ensure public safety by mitigating human exposure to tritium.

**Keywords:** tritium concentration; groundwater; shallow wells; natural radionuclide; deep wells; scintillation counter

## 1. Introduction

Tritium, an isotope of hydrogen, is naturally present in the upper atmosphere through the interaction of cosmic rays and nitrogen molecules [1]. Artificial tritium can be produced during nuclear explosions or as a byproduct in nuclear reactors, often reacting with oxygen to form water. Once tritium is produced naturally or artificially, tritium can enter the hydrological cycle and might be incorporated into groundwater sources, soil, and plants [2,3]. Tritium in groundwater can pose a health risk by ingesting water, inhaling as gas from the air during washing and bathing, or being absorbed through the skin. Exposure to tritium can damage soft tissues and organs, leading to various health issues [4]. The international agency for research on cancer (IARC), which operates under the World Health Organization (WHO), has categorized tritium as a cancer-causing agent [5,6] and has established an acceptable limit of tritium in water. The reference concentration level for drinking water has been established at 11.1 Bq/L by both the U.S. Environmental Protection Agency (USEPA) and the European Atomic Energy Community (EAEC) [7]. In contrast, the WHO recommends a safe level of 100 Bq/L and warns that if the tritium level is exceeded, the water needs to undergo treatment until the concentration is lowered to 100 Bq/L.

Due to the potential health risks associated with tritium, assessing water has become crucial for researchers worldwide. Many studies have been conducted on the tritium concentration in groundwater and the corresponding health risk assessment. For example, precipitation samples collected in Australia in 1963 showed tritium concentrations of 1355 Bq/L [8]. During the same year, Ottawa (Canada) and Vienna (Austria) observed an average yearly tritium concentration in precipitation samples of about 24,000 Bq/L

and 27,000 Bq/L, respectively, as a consequence of thermonuclear tests. In Thailand, tritium measurements began in 1968, and rainwater samples collected in Bangkok exhibited a tritium concentration of approximately 390 Bq/L [9,10]. It has been confirmed that atmospheric thermonuclear tests are the main contributors to the elevation of tritium levels in the environment. In one study, the areas are closer to the present searched area; the groundwater wells are classified as a mix of submodern and modern water, and the groundwater was not drinkable but could be used for domestic purposes [11]. Notably, the acceptable levels of tritium in groundwater can vary depending on the specific conditions and regulations of a given location. Sometimes even low tritium concentrations may be unacceptable, mainly if detected in sensitive areas near drinking water sources or densely populated regions. Therefore, monitoring tritium levels in groundwater is crucial to ensuring the safety of human health.

In earlier research, radon concentration in groundwater was evaluated using a liquid scintillation counter and alpha spectrometry [12–14]. The findings revealed that radon concentrations in groundwater are far below the limits established by international organizations. Moreover, a comparison was made between the results obtained in the Gulf countries. The present study aims to estimate the tritium concentration in spiked, rain- and groundwater from different shallow and deep wells used for drinking, domestic, and agricultural purposes. The results are compared with two international laboratories measuring the tritium concentration in water commercially. To the best of our knowledge, this is the first attempt to measure tritium levels in northeastern Saudi Arabia.

## 2. Materials and Methods

### 2.1. Samples

Five types of water samples were used in this study, each containing different levels of tritium, as tabulated in Table 1. The first type of sample comprised two standards provided by Hidex, London, UK, with very high levels of tritium and carbon-14. The second type included five reference samples with a known (high to low) tritium concentration. The reference samples were prepared by diluting a certified standard solution provided by Eckert & Zeigler (Catalog Number: P700723, Lot Number: 1676-44) with tritium-free water, following controlled conditions. The final concentration of the tritium level was estimated using the dilution factor. The third type includes rainwater samples collected at different times in winter. The fourth type is groundwater samples directly collected in thirteen different locations, assumed to have very low tritium levels. Groundwater samples were collected directly from monitoring wells, carefully preserved in glass bottles, and promptly sealed upon collection. Measurements were typically conducted within a month of sample collection for higher accuracy unless specified in the text. Finally, a single blank sample with almost zero tritium (dead water) was used for background measurement.

**Table 1.** Samples used in the present study.

| Sl No. | Sample Type | Total Sample | Tritium Level |
| --- | --- | --- | --- |
| Type 1 | Standard sample | 2 | Very high |
| Type 2 | Reference water | 5 | High to low |
| Type 3 | Rainwater | 4 | Low |
| Type 4 | Groundwater | 13 | Very low |
| Type 5 | Background water | 1 | ~Zero |

The tritium concentrations in the reference water, rainwater, and groundwater were estimated by following the diagram as shown in Figure 1.

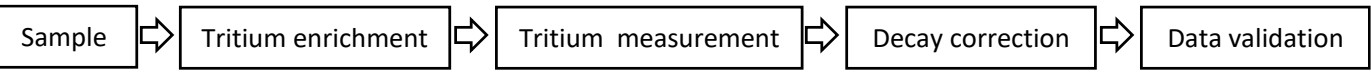

**Figure 1.** Schematic diagram of the procedure for tritium concentration calculation.

## 2.2. Tritium Enrichment

Tritium levels in water samples are low; in most cases, low tritium levels cannot be detected by a standard liquid scintillation counter (LSC). In order to determine the tritium concentration it is essential to enhance the presence of low-level tritium in water by a known factor. This enhancement allows the scintillation counter to be capable of detecting it. Therefore, electrolytic enrichment was performed as instructed by the International Atomic Energy Agency (IAEA). Finally, the scintillation counter measured the tritium activity concentration of the enriched samples.

The initial step of the electrolysis process involves the primary distillation of groundwater by elevating its temperature to the point of boiling, which triggers the creation of steam. This vapor is subsequently gathered, cooled, and transformed back into its liquid state. Throughout this sequence, pollutants and undesired substances are extracted. The distillation process removed dissolved ions of chloride, sulfate, calcium, carbonate, magnesium, sodium, and potassium and dissolved solids commonly found in groundwater. The absence of these ions and salts was confirmed using a YSI 9500 photometer and a Horiba digital meter.

A batch of electrolytic cells with a maximum capacity of 800 mL water was used with stainless steel anodes and mild steel cathodes. The electrolysis process typically takes one to two weeks to enrich 500–800 mL of water, and up to nine cells can be operated simultaneously. After the electrolytic process, the final volume of the water sample was collected from the electrolytic cells. The calculated enrichment factor was approximately ten to forty times, with an estimated error range of 0.5 to 2.0. Details of the electrolysis enrichment process, enrichment time, enrichment factors, and data validation are provided in a previous study [15].

Neutralizing the alkalinity of the solution is crucial because it can interfere with the electrolytic process. An acidic substance is added to the solution to neutralize the alkalinity. A pH of 7 was set as a target when neutralizing an alkaline solution. Once the electrolytic process was completed, the electrolytes in the samples were removed by final distillation. Following the last distillation step, the water samples were once more examined to ensure the absence of electrolytes.

## 2.3. Tritium Measurement

Upon completion of the last distillation phase, 10 mL of the water sample was combined with an equal amount of scintillation cocktail (Aqualight+, Hidex, London, UK). This mixture serves to transform the energy released during tritium decay into light flashes. Following this, the samples were stored in a cool and dark setting for at least 4 h before undergoing measurements using a liquid scintillation counter (LSC), a measure taken to prevent any luminescence interference. The evaluation of tritium content was conducted using a Hidex 300 SL liquid scintillation counter, which functions on the principle of triple-to-double coincidence ratio (TDCR). Figure 2 provides photographs of the measuring device, highlighting its scintillation principle. The device consists of three photomultiplier tubes positioned at a 120-degree angle to each other (Figure 2c,d). With the help of these three photomultipliers, the triple-to-double coincidence ratio is measured, allowing the determination of the counting efficiency of an unknown sample without requiring a standard source.

For each sample, analysis was performed five times, with a counting time of 200 min. The experiments were conducted at a constant temperature, while the operating area was selected to minimize gamma radiation and eliminate light. The instrument was checked for drift before measuring the activity. The calibrations were performed using a standard source to monitor the instrument's performance. Additionally, the background radiation levels were continuously monitored. The super low-level option was employed by selecting the shield in the Mikrowin software that controlled the LSC. The shield can be used with various counting modes, including standard, low, alpha-beta separation, and triple mode

ROIs (regions of interest). It is important to mention that the impact can be overlooked when the measured activities are roughly 1000 times greater than the background.

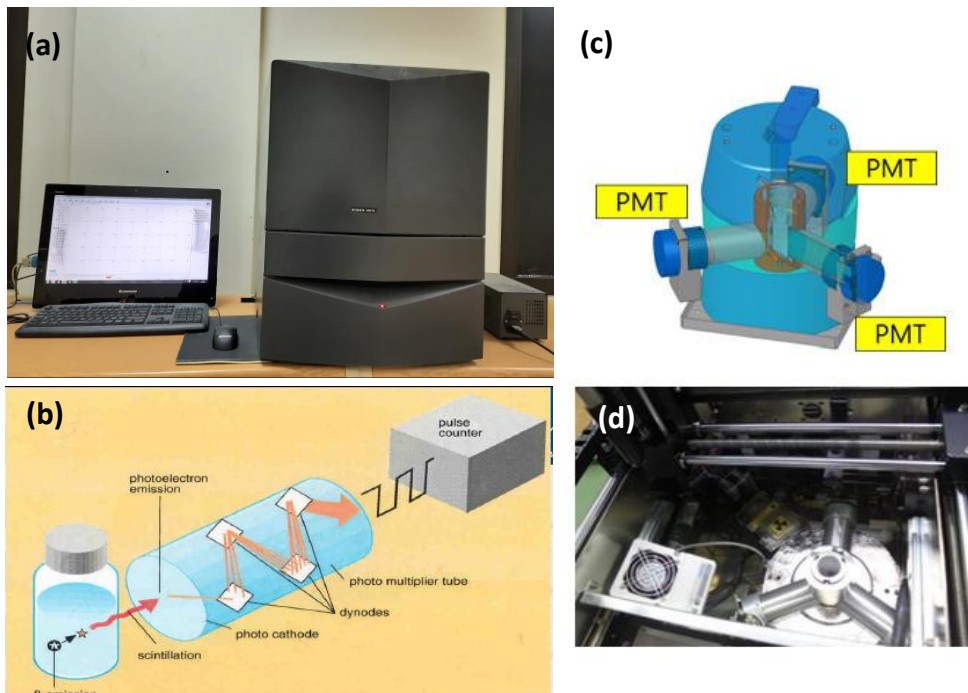

**Figure 2.** (**a**) The front view of the tritium-measuring device (Hidex 300SL), (**b**) a schematic representation of the scintillation principle, (**c**) the diagram of the photomultiplier, and (**d**) a photograph of the detector (Hidex 300SL).

The Hidex 300 SL is equipped with a feature that discharges static charge from the sample just before measurement to prevent the influence of static electricity. The discharge process extends the sample processing time by 1 to 2 s. To achieve accurate determination of the activity of an unknown sample, it is crucial to have precise knowledge of both the efficiency and background. To ensure this, five additional samples of Type 2 were prepared using the same chemistry and conditions as the unknown samples. These additional reference samples included a 10 mL aqueous sample mixed with a 10 mL water-soluble cocktail.

### *2.4. Mathematical Formulation*

2.4.1. Efficiency of the Scintillation Counter

A standard source (Type 1 samples provided by Hidex, UK) was used for efficiency measurement, and the measurement was carried out using identical conditions as those applied to the unknown samples. The tritium measurement efficiency, $\varepsilon$, was obtained from the net count rate of the standard sample, $R_n$ (s$^{-1}$); the known activity concentration, $c$ (Bq/L); and the sample volume, $m_M$ as

$$\varepsilon = \frac{R_n}{c \times m_M} \tag{1}$$

The light flashes are measured in counts per minute (CPM) using a liquid scintillation counter (Hidex 300 SL). Subsequently, the average of the five trials was calculated. Standards and zero-value samples (Type 5 blanks) were also analyzed under comparable experimental conditions to assess the data.

### 2.4.2. Tritium Concentration

After the enrichment step, the tritium activity concentration, '$c$', at the sampling time was determined using the following equation:

$$c = \frac{m_n \times \rho \times f_A}{\varepsilon \times \eta_{A,i} \times m_V \times m_M} \times R_n = \frac{m_n \times \rho \times e^{\lambda \cdot t_A}}{\varepsilon \times \eta_{A,i} \times m_V \times m_M} \times R_n = \varphi \times R_n \tag{2}$$

For both standard and background samples, where the enrichment step is no longer needed, Equation (2) can be rewritten as

$$c = \frac{\rho \times e^{\lambda \cdot t_A}}{\varepsilon \times m_M} \times R_n \tag{3}$$

where

$c$: activity concentration of tritium in Bq m$^{-3}$;
$f_A$: correction factor for the decay;
$\lambda$: decay constant of tritium in s$^{-1}$;
$t_A$: time between sampling and beginning of the measurement in s;
$R_n$: net count rate of the counting source in s$^{-1}$; with $R_n = R_g - R_o$
$\varepsilon$: detection efficiency in Bq$^{-1}$ s$^{-1}$;
$\rho$: density of the groundwater in kg m$^{-3}$;
$m_n$: mass of the solution remaining in the electrolysis cell after electrolysis, in g;
$m_V$: mass of the solution filled into the electrolysis cell before electrolysis, in g;
$m_M$: mass of distillate in the scintillation vial, in g;
$\varphi$: procedural calibration factor Bq s m$^{-3}$.

The conversion of mass was performed by considering the water volume, where 1 m$^3$ is equivalent to 1000 kg. The overall gross count rate, '$R_g$', was computed from the individual gross count rates, '$R_{g,j}$', using the following equation:

$$R_g = \frac{\sum_1^j R_{g,j} \times t_{m,j}}{\sum_1^j t_{m,j}} = \frac{\sum_1^j R_{g,j} \times t_{m,j}}{t_{total}} \tag{4}$$

where

$R_{g,j}$: gross count rate of the single measurement, $j$, of the counting source, in s$^{-1}$;
$t_{m,j}$: duration of the single measurement, $j$, in s;
$t_{tot}$: sum of the durations of a single measurement in s.

### 2.4.3. Decision Threshold and Detection Limit

For the measurement of the activity of the sample, the counting time is crucial. The counting duration assesses the decision threshold and the minimum detectable activity. Equations (6) and (7) were applied for the computation of the decision threshold and minimum detection limit.

$$c^* = k_{1-\alpha} \times \varphi \times \sqrt{\left[ R_o \times \left( \frac{1}{t_{total}} + \frac{1}{t_o} \right) \right]} \tag{5}$$

$$c^{\#} = c^* \times k_{1-\beta} \times \sqrt{\left( c^{\#2} \cdot u_{rel}^2(\varphi) \right) + \varphi^2 \cdot \left( \frac{c^{\#}}{t_{total} \cdot \varphi} + \frac{R_o}{t_{total}} + \frac{R_o}{t_o} \right)} \tag{6}$$

where
$c^*$: decision threshold in Bq m$^{-3}$;
$c^{\#}$: detection limit in Bq m$^{-3}$;
$k_{1-\alpha}$: quantile of the normal distribution for $\alpha = 0.0014$;

$k_{1-\beta}$: quantile of the normal distribution for $\beta = 0.05$.

Once the auxiliary values, $\psi$ and $\theta$, have been introduced, the detection limit, $c^{\#}$, can be computed using the subsequent equation:

$$c^{\#} = \frac{c^* \cdot \psi}{\theta} \cdot \left[ 1 + \sqrt{1 - \frac{\theta}{\psi^2} \cdot \left( 1 - \frac{k_{1-\beta}^2}{k_{1-\alpha}^2} \right)} \right] \quad (7)$$

where

$$\theta = 1 - k_{1-\beta}^2 \cdot u_{rel}^2(\varphi) \quad (8)$$

$$\psi = 1 + \frac{k_{1-\beta}^2}{2 \cdot c^*} \cdot \varphi \cdot \frac{1}{t_{total}} \quad (9)$$

### 2.4.4. Standard Uncertainty

The standard uncertainty of the analysis comprises uncertainties related to counting statistics, calibration of the liquid scintillation counter, tritium yield of the electrolytic cell, and the sample volume considering density and mass. However, uncertainties for the measurement time and decay constant were excluded. Thus, the standard error of the activity concentration, '$u(c)$', was calculated using the following equation:

$$u(c) = c \cdot \sqrt{\left[ \frac{1}{(R_g - R_o)^2} \times \left( \frac{R_g}{t_{total}} + \frac{R_o}{t_o} \right) \right] + u_{rel}^2(\varphi)} \quad (10)$$

In the absence of electrolysis, the relative standard uncertainties for the tritium yield and masses will be eliminated both before and after the process.

$$u_{rel}^2(\varphi) = u_{rel}^2(f_A) + u_{rel}^2(m_n) + u_{rel}^2(m_V) + u_{rel}^2(m_M) + u_{rel}^2(\varepsilon) + u_{rel}^2(\eta_{A,i}) + u_{rel}^2(\rho) \quad (11)$$

where

$t_o$: is the sum of the time for a single background measurement, $t_{o,j}$, in s;
$u_{rel}(f_A)$: relative standard uncertainty of the correction factor for the decay;
$u_{rel}(\eta_{A,i})$: relative standard uncertainty of the tritium yield of electrolysis cell, $i$;
$u_{rel}(m_n)$: relative standard uncertainty of the mass of solution remaining in the electrolysis cell before electrolysis;
$u_{rel}(m_v)$: relative standard uncertainty of the mass of solution filled into the electrolysis cell before electrolysis;
$u_{rel}(m_M)$: relative standard uncertainty of the mass of distillate in the scintillation vial;
$u_{rel}(\rho)$: relative standard uncertainty of the density of water;
$u_{rel}(\varepsilon)$: relative standard uncertainty of the detection efficiency.

The calculated uncertainty for the three tested labs was different. Here, it is important to note that one tritium unit (1 TU) is equal to 0.118 Bq/kg, and all the calculated units were converted in TU.

### 2.4.5. Decay Correction

When there is a considerable time gap between sample collection and measurement, it becomes necessary to correct the measured tritium concentration due to sample decay. If the sample is collected at an earlier time but analyzed later, the tritium concentration will naturally decrease during the intervening delay. Hence, it is crucial to apply a correction for the delay time between sampling and measuring. The decay correction factor (DCF) is a

simple exponential function characterized by a time constant of 17.745 years. Its calculation is based on the subsequent formula:

$$DCF = e^{(T/17.745)} \tag{12}$$

In the equation, '$T$' represents the time duration (in years) between the sample collection and the counting process using the LSC instrument, and the value of 17.745 years corresponds to the average lifetime of the tritium ($^3$H) atom. Tritium has a half-life of 12.3 years, which is divided by the natural logarithm of 2 to obtain this mean lifetime value [16].

### 2.5. Data Validation by Comparing Results from Other Labs

To authenticate the collected data, identical groundwater samples were sent to two commercial tritium laboratories: "Isodetect Womweltmonitoring GmbH" in Germany and the University of Miami in the USA. Tritium analysis at the Isodetect laboratory was conducted using Quantulus W1220, while at the University of Miami, Tri-carb 3170TR/SL was employed for the analysis. Similar experimental conditions were followed in both laboratories for the sampling, distillation, and electrolysis processes. Five analyses were conducted on each sample, with each analysis lasting 200 min. Samples with an initial volume of 400 mL are electrolyzed to a remaining volume of 20 mL, corresponding to a 20-fold tritium enrichment in both laboratories.

### 3. Results

#### 3.1. Efficiency and Background Calibration Using a Standard Sample

The performance of the scintillation counter and estimation of the tritium concentration requires calibration and efficiency calculations. The performance of LSC was periodically monitored using standard samples (Type 1 and Type 5). The calibration process involved certified carbon-14 ($^{14}$C) and tritium ($^3$H) sources and activity-free water. The calibration results are illustrated in Figure 3. Over the course of the past eight years, the activity-free water consistently showed a stable reading with negligible experimental error, indicating a constant radiation level in both the activity-free solution and the surrounding environment (Figure 3a). Due to its extended half-life (around 5715 ± 40 years) [17], the count per minute (CPM) of the standard $^{14}$C source remained relatively stable without any significant variations.

In contrast, the count from the standard tritium source ($^3$H) displayed a gradual exponential decrease over the past eight years since the device's initialization, reaching approximately 0.6162 times the initial count (Figure 3b). Upon scrutinizing the exponential curve and employing Equation (1), a preliminary assessment indicates a half-life of approximately 11.75 years for $^3$H, closely mirroring the reported 12.3 years in the literature [17]. Conversely, the count per minute (CPM) of the conventional $^{14}$C source exhibited consistent levels, attributed to its notably lengthier half-life of around 5715 ± 40 years. The steady background count and sustained $^{14}$C count over time serve as indicators of the remarkable operational prowess of the liquid scintillation instrument. It is important to note that the effect of the background count can be disregarded when the measured value is approximately 1000 times higher than the background count.

#### 3.2. TDCR with Efficiency and Repeating Cycle

The Hidex 300SL liquid scintillation counter utilizes the triple-to-double coincidence ratio (TDCR) method to estimate the tritium concentration in groundwater samples. The TDCR method is specifically designed for measuring the activity of pure alpha- and beta-emitters. It involves calculating the detection efficiency by employing a statistical and physical model to analyze the photon distribution emitted by the scintillating source [18]. The Hidex 300SL device's detector is fitted with three photomultipliers, allowing the measurement of the triple-to-double coincidence ratio. This method enables the determination

of the counting efficiency of an unknown sample without requiring a standard source [19]. The TDCR value represents the ratio between the count rate of triple coincidences and double coincidences. As detected events increase, the TDCR value approaches higher efficiency [20]. Figure 4a illustrates the TDCR values plotted against the efficiency, displaying a linear relationship characteristic of the TDCR method utilized by the Hidex 300SL device.

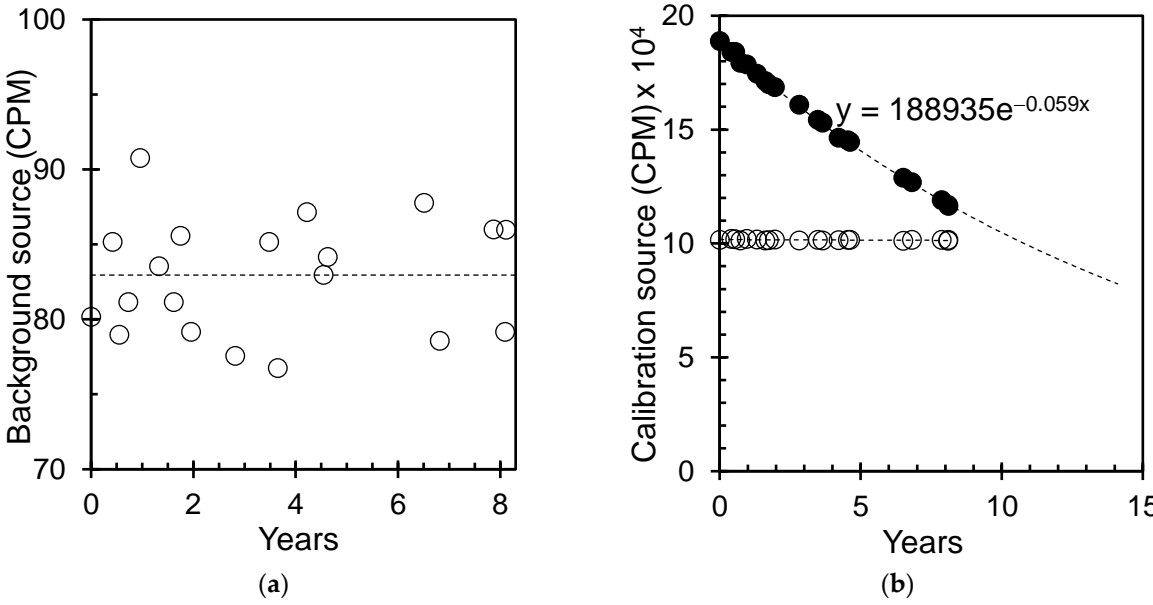

**Figure 3.** An eight-year record for the calibration source, including (**a**) the Type 5 sample of activity-free water for background, (**b**) the Type 1 sample featuring certified sources of $^{14}C$ (shown as open circles) and $^3H$ (solid circles). The flashes are measured in counts per minute (CPM). The dashed curve for $^3H$ extends to visualize the exponential dependence with time.

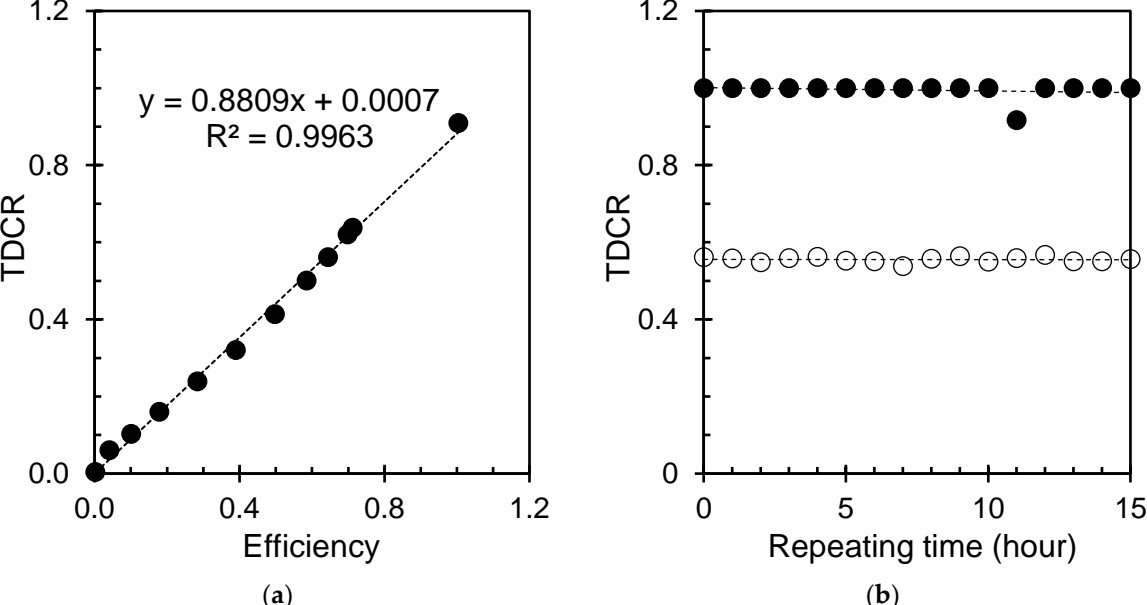

**Figure 4.** TDCR as a function of (**a**) efficiency and (**b**) repeating run (converted into time in hours) for a typical sample 12 of Type 4 for $\alpha$-radiation (solid circle) and $\beta$-radiation (open circle).

Additionally, it was found that the TDCR value for alpha radiation was higher than that for beta radiation, as shown in Figure 4b. Conversely, no notable variation in TDCR

values was observed for repeated runs. These findings suggest that the TDCR method allows for the measurement of tritium activity concentration even in the case of multiple repeated runs.

### 3.3. Activity-Dependent Efficiency

The triple-to-double coincidence ratio (TDCR) method is widely employed to measure small quantities of radioactivity in diverse sample types. In this method, the sample is combined with a liquid scintillator, a substance that emits light when stimulated by radiation. The intensity of the emitted light is directly proportional to the level of radioactivity present in the sample.

The TDCR efficiency can be influenced by the activity concentration of the sample. A known activity of the spiked sample was diluted, and the activity concentration of each sample was measured along with its corresponding TDCR efficiency. A curve was obtained by plotting the TDCR efficiency against the known activity concentration, as depicted in Figure 5. The graph illustrates that the TDCR efficiency tends to be lower at lower activity concentrations, whereas at higher activity concentrations, the TDCR efficiency increases. This variability in TDCR efficiency with activity concentration arises from the finite resolution time of the detector and statistical fluctuations in the radiation. A similar pattern was observed in a study by Broda et al. [21]. As a result, before the experiment, the TDCR was calibrated for each sample to accurately measure its activity concentration.

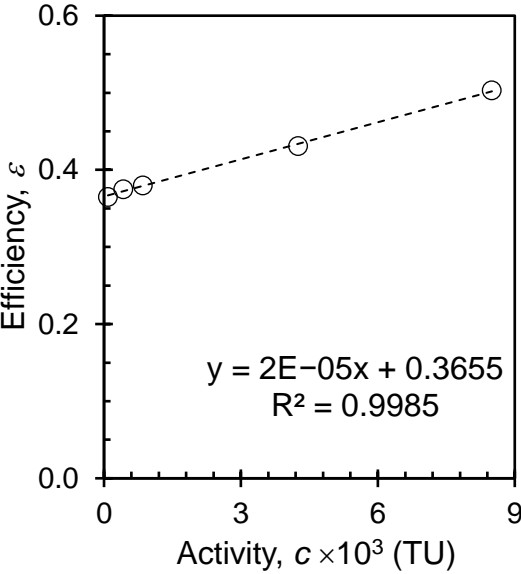

**Figure 5.** Efficiency variation as a function of activity concentration of Type 2 reference samples. The reference samples were prepared by diluting a standard sample with activity-free water.

### 3.4. Counting Time-Dependent Decision Threshold and Detection Limit

A reliable statistical outcome from the liquid scintillation counter necessitates an adequate amount of time. The detection limit signifies the minimal radioactivity level that the counter can identify while upholding a specified confidence level. The duration of counting needed to attain the desired statistical confidence level hinges on several factors, including the counter's detection limit, sample size, sample activity, and the desired confidence level itself. Hence, optimizing the counter's detection limit and tailoring the counting time are pivotal in achieving the preferred statistical confidence level when utilizing a liquid scintillation counter, such as the Hidex 300 SL.

Figure 6 shows the variation in minimum detection activity, MDA, and decision threshold as a function of counting time. Figure 6 demonstrates that counting time significantly impacts the detection limit, with longer counting times resulting in lower detection limits. A higher counting time lowers the detectable activity and decision threshold. In the case of

groundwater samples, tritium levels are typically very low and require a longer counting time to achieve a lower detection limit. In this study, a counting time of 200 min with five repetitions was chosen, resulting in a detection limit of approximately 0.26 TU. A comparable pattern of the detection limit was noted in relation to the counting time by Feng et al. [22]. They estimated the minimum detectable activity (MDA) of the LSC-LB7 system using 100-mL vials to be 1.525 TU with a continuous counting time of 3600 min, whereas the 20-mL vials contained 6 TU.

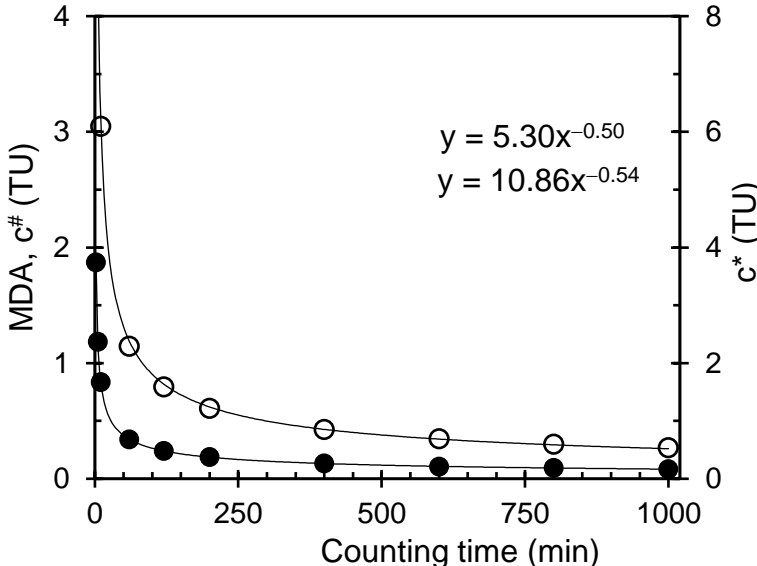

**Figure 6.** Counting time-dependent minimum detection activity, $c^{\#}$ (TU) (open circle), and decision threshold, $c^{*}$ (TU) (solid circle), for a typical sample Gr-12 of the Type 4 sample.

### 3.5. Decay Correction Factor for Sampling Time

Decay correction is vital as the radioactive material continuously decreases due to decay. With a time gap between sample collection and LSC measurement, the decay correction can adjust measurements to reflect decay since material acquisition. The correction ensures accuracy, particularly for long-lived radionuclides or extended measurements.

Figure 7 illustrates the decay correction factor as a function of time (represented by a dotted line). The plot shows that if the samples are appropriately sealed, stored, and counted by the LSC within a year, it does not show a significant correction. However, if the sample is stored for more extended periods, the decay correction factor (DCF) exhibits exponential growth over time. In this study, most samples were counted within a few months of sample collection, except for a few (represented by open circles in Figure 7) collected from other cities and analyzed after being sent to Isodetect, Germany and Miami lab, USA within a year. A proper correction using the DCF equation was applied, and the corrected values are presented in Table 2.

**Table 2.** Activity concentration of tritium in the reference samples.

| Sample ID | Spiked Activity (TU) | ±Error (TU) | Measured Activity (TU) | ±Error (TU) | Z |
|-----------|----------------------|-------------|------------------------|-------------|--------|
| Ref-1 | 8500 | 604.4 | 8760 | 560.9 | −0.346 |
| Ref-2 | 4250 | 182.7 | 4124 | 264.1 | 0.352 |
| Ref-3 | 850 | 166.2 | 820 | 52.42 | 0.160 |
| Ref-4 | 425 | 133.4 | 395 | 25.29 | 0.213 |
| Ref-5 | 85 | 12.29 | 89.5 | 5.740 | −0.352 |

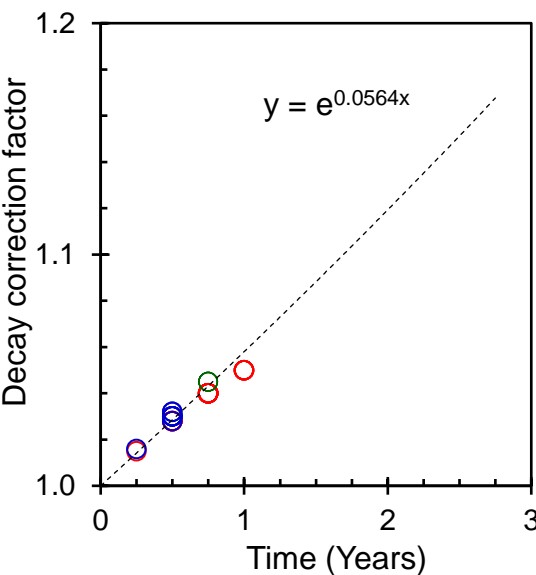

**Figure 7.** Time dependent DCF for the analyzed sample in Saudi Arabia (Lab 1, red circle), the Isodetect lab in Germany (Lab 2, green circle), and the University of Miami lab in the United States (Lab 3, blue circle). The dotted line is calculated from the decay equation.

*3.6. Tritium Activity Concentration*

3.6.1. Reference Samples

Validation with reference samples is a critical practice to confirm the accuracy of the measurement procedure. It involves comparing the results of an experimental or analytical method with known reference samples to ensure the method's accuracy, reliability, and reproducibility. By validating a method using reference samples, one can gain confidence in the accuracy, precision, and reliability of the obtained results, which is crucial for making informed decisions, drawing meaningful conclusions, and ensuring the quality of scientific or analytical processes.

In order to validate the measurement procedure's accuracy, the tritium concentration in the reference samples was assessed using LSC. The Z score, a statistical tool, was employed to assess the degree of deviation between actual and expected values. This calculation considers the number of measurements taken, the measurements' standard deviation, and the discrepancy between measured and expected values. The resultant Z score value should ideally range between $-2$ and 2 for satisfactory quality. The specific activity of the reference sample with its corresponding Z score is detailed in Table 2. The values within the table, ranging between $-0.3$ and 0.4, demonstrate that the measured values are well within an acceptable range of the anticipated values, even considering measurement uncertainties. This reaffirms that the measurement system is both accurate and precise.

3.6.2. Rainwater Samples

Monitoring tritium levels in rainwater is part of environmental surveillance efforts to assess the potential impacts of radioactive releases from rain. Tritium concentrations in rainwater can vary depending on location, atmospheric conditions, and human activities. Typically, the tritium levels in rainwater are low. It is worth noting that tritium is a weak beta emitter, and its penetration power is limited. External exposure to tritium through contact with rainwater is generally not a significant concern. The primary concern arises from internal exposure if tritium-contaminated rainwater is ingested or used as a water source for drinking.

The tritium level in rainwater was estimated and tabulated in Table 3. The tritium levels are in the range of 2.8–3.5 TU with a significant error. The tritium level in rainwater can vary depending on several factors, such as geographical location, atmospheric conditions,

and proximity to tritium sources. Generally, tritium concentrations in rainwater are very low. In areas far from significant tritium sources, the tritium concentration in rainwater typically ranges from a few to tens of tritium units. These levels are considered safe and do not pose significant health risks to the general population. However, the present study found the tritium level of rainwater to be safe.

**Table 3.** Tritium activity concentration of the rainwater sample.

| Sample ID | Tritium Activity (TU) | ±Uncertainty (TU) |
|---|---|---|
| Rain-1 | 3.37 | 0.23 |
| Rain-2 | 3.08 | 0.21 |
| Rain-3 | 3.48 | 0.24 |
| Rain-4 | 2.78 | 0.20 |

### 3.6.3. Groundwater Samples

The tritium concentrations in groundwater samples collected from the northeastern part of Saudi Arabia are presented in Table 4. The concentrations ranged from 0.3 to 0.8 TU, with a significant standard deviation. The highest tritium concentration was observed in the northernmost area, specifically in the lower valley of the surveyed region. Overall, the surveyed site exhibited very low levels of tritium.

**Table 4.** Tritium activity concentrations of the groundwater samples collected from different wells.

| Sample ID | Depth of Wells | Tritium Activity (TU) Lab 1 | ±Uncertainty (TU) |
|---|---|---|---|
| Gr-01 | Deep | 0.47 | 0.07 |
| Gr-02 | Deep | 0.58 | 0.08 |
| Gr-03 | Deep | 0.50 | 0.07 |
| Gr-04 | Deep | <MDA | - |
| Gr-05 | Shallow | 0.40 | 0.06 |
| Gr-06 | Shallow | 0.32 | 0.06 |
| Gr-07 | Shallow | 0.42 | 0.07 |
| Gr-08 | Deep | <MDA | - |
| Gr-09 | Deep | <MDA | - |
| Gr-10 | Shallow | 0.44 | 0.07 |
| Gr-11 | Deep | <MDA | - |
| Gr-12 | Deep | <MDA | - |
| Gr-13 | Shallow | 0.78 | 0.09 |

Different national and international entities have developed worldwide guidelines for tritium levels. As an example, the United States Environmental Protection Agency (USEPA) has established a threshold of 11.1 Bq/L as the limit for tritium concentration in groundwater [23]. The United Nations Scientific Committee on the Effects of Atomic Radiation (UNSCEAR) has defined a safe limit of 40 Bq/L in their published report [24], while the European Commission (EC) and World Health Organization (WHO) consider 100 Bq/L as the action limit [7,25]. It is worth noting that all the samples analyzed in this study had tritium levels far below these limits.

Additionally, approximately 90% of the samples had tritium concentrations significantly lower than those reported in previous studies. The estimated tritium concentrations in this study were lower than those in other regions in Saudi Arabia, ranging from 1.0 to 80 TU [26–28].

### 3.7. Data Validation

Data validation plays a critical role in the measurement process. It involves verifying the accuracy and reliability of the estimated data to ensure its validity for further analysis. The validation process involves various steps, such as error checking and comparison with

known values. Error checking involves detecting anomalies or outliers that could impact result accuracy, often using regression analysis and statistical methods.

A comparison of results obtained from multiple laboratories was made by considering potential errors or discrepancies resulting from testing methods and equipment variations as shown in Figure 8. Standardizing protocols, procedures, and testing standards across laboratories is necessary to minimize variations and enable proper comparison. Calculating measurement uncertainty for each laboratory is crucial for comparing results, as it provides a measure of confidence in the accuracy of the measurements. This approach helps identify significant differences in precision or accuracy between laboratories and determines if they fall within acceptable ranges.

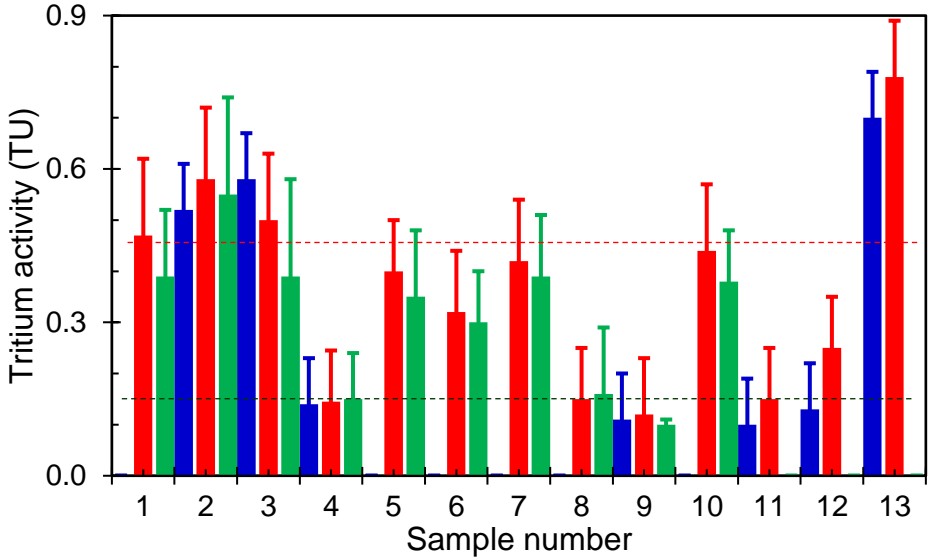

**Figure 8.** A comparison of tritium concentration in groundwater samples was estimated in various analytical laboratories: lab 1 in Saudi Arabia (red); lab 2, Isodetect lab in Germany (green); and lab 3, the University of Miami lab in the United States (blue).

An alternative approach to data analysis involves conducting statistical examinations to identify significant variations or patterns within the results. This may include calculating statistical metrics, such as the average, standard deviation, and coefficient of variation for each dataset and comparing these values to detect any noticeable differences or trends in the data.

The data validation process involves comparing the obtained data with those from commercial laboratories to ensure their accuracy and reliability. This comparison serves as confirmation of the results. Two international commercial tritium laboratories, namely, Isodetect Umwelmonitoring GmbH in Germany and the University of Miami in the USA, were involved in this validation process. The statistical relationship between the results obtained from different laboratories is illustrated in Figure 9. When comparing the tritium data from our laboratory with those from the other laboratory for the same sample, the results indicate that the data estimated in our laboratory are valid, with negligible error. The tritium concentration, determined through liquid scintillation counting, has been compiled in Table 4 for the two distinct commercial laboratories.

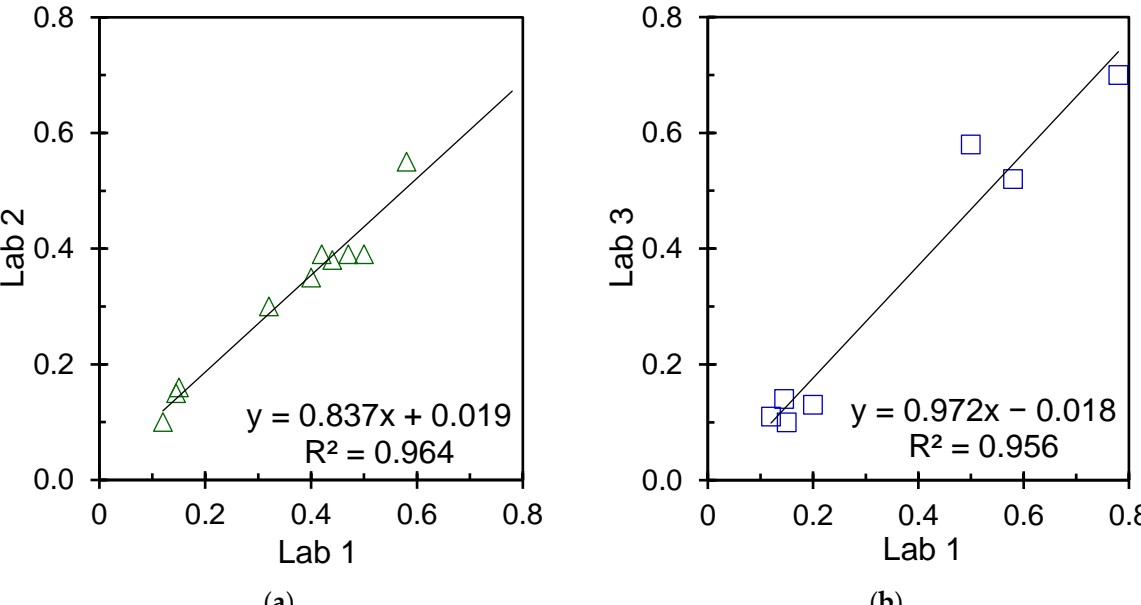

**Figure 9.** A comparison of tritium concentration between the analytical laboratories: (**a**) lab 1 (UHB, Saudi Arabia) and lab 2 (Isodetect lab, Germany), and (**b**) lab 1 (UHB, Saudi Arabia) and lab 3 (the University of Miami lab, United States).

## 4. Conclusions

The tritium concentration in various types of water was estimated using a liquid scintillation counter in the dry river of Hafr Al Batin, Saudi Arabia. The water samples were enriched ten- to forty-fold by the electrolysis process. Five reference water samples confirm the reliability of the estimation method. The estimated tritium concentrations for rainwater and groundwater in the dry river of Al Batin ranges from 0.3 to 3.5 TU, which falls below the safety limits established by international organizations. Shallow wells in the most northern area, the lower valley, exhibited relatively high tritium concentrations. In contrast, wells located in the hillside areas demonstrated comparatively lower concentrations. The relatively low levels of tritium may be attributed to the geological composition of the rocks and sediments through which the water flows. Two reputable international tritium laboratories further verified the estimated concentrations, confirming their accuracy with minimal error. These findings indicate that tritium exposure from groundwater in the study areas is within the acceptable limits of international safety standards.

**Funding:** This research was funded by the Institutional Funding Project (IFP-A-202-1-3) from the Ministry of Education, Saudi Arabia.

**Data Availability Statement:** The data presented in this study are available in this article.

**Acknowledgments:** The authors express their gratitude to the Deanship of Scientific Research, University of Hafr Al Batin, for providing the experimental facilities. Furthermore, the work receives financial support from the Institutional Funding Project (IFP-A-202-1-3) granted by the Ministry of Education, Saudi Arabia. The author also thanks Isodetect Umwelmonitoring GmbH, Germany and the University of Miami, USA for prompt help validating the data.

**Conflicts of Interest:** The author declares no conflict of interest. The funders had no role in the study's design; in the collection, analyses, or interpretation of data; in the writing of the manuscript; or in the decision to publish the results.

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
