# Peer review of "Estimation of Tritium Concentration in the Rain- and Groundwater in the Dry River of Hafr Al Batin, Saudi Arabia"

_2300-7575, doi:10.3390/limnolrev23020006_

Round 1

Reviewer 1 Report

The article is interesting, well-structured and worth publishing. However, a few issues require clarification.

The main problem is the question of the scientific novelty of the research. The authors indicate the elements of scientific novelty in the context of local research. However, on the basis of literature data, should be indicated how described studies bring new elements to the broader knowledge.

The experimental part of the research has been thoroughly and well documented and does not raise any doubts. Similarly, the discussion and conclusion are based on the results and clearly indicate environmental dependencies.

Reviewer 2 Report

This is an article on the estimation of tritium in different type of water samples around Hafr Al Batin area.

It is a detailed description of the measurement technique and equipment, but I am not sure if the study itself is enough for an article. It includes many definitions and basic things. Study with not too many samples, although the area, extension, interest, the location of the samples, etc. is not detailed either. This paper has few novelties, except for the area in which the analyzes have been carried out, so more emphasis should be placed on this section and on the conclusions of the results.

The following points should also be considered:

Abstract: TU is mentioned but the meaning should be explained here.

Introduction: 

- Line 53: please, explain better (not drinkable?)

- Line 59: radon in water?

Materials and Methods

Line 110: same error 0.5 for ten as for forty?

Line 166: Text repeated from Line 132

Line 185: net count rate: counts per second, c · s-1 vs s-1 only?

Line 253: Check converted “in” TU

Line 259: Rewrite based on the sampling time to better express the delay time between sampling and measuring.

2.4.5 Decay correction: Is there a limitation before apply this correction? If the value measured is MDA?

Lines 273-275: Rewrite to better express laboratories devices

Figure 3.a: adapt the scale of the y-axis (70-100 cpm?)

Line 337: Text repeated from Line 316

Lines 341-343: It is Figure 5 instead of Figure 6, and the “graphical analysis” is wrong.

Figure 6: representing one is enough. One quantity is proportional to the other and inversely proportional to the square root of time.

Lines 361-363: Obvious

Lines 372-379: very obvious paragraph, please summarize.

Figure 7: It is not clear the usefulness of this graph. Why the red circles deviate from the trend line?

Tables 2, 3 and 4: Check if "Error" should be replaced by "Uncertainty"

Line 410: Revise z-score values in the text with Table 2

Line 425: 2.78-3.37 or 2.8-3.4

Line 434: Revise concentration values in the text with Table 4

Round 2

Reviewer 2 Report

I would like to thank the author for reviewing and accepting almost all of my comments and for providing answers and clarifications on all of them.

I have no further comment other than that I find the article ready for publication in its present form.

Kind regards.